# Characterization of a Lytic Bacteriophage and Demonstration of Its Combined Lytic Effect with a K2 Depolymerase on the Hypervirulent *Klebsiella pneumoniae* Strain 52145

**DOI:** 10.3390/microorganisms11030669

**Published:** 2023-03-06

**Authors:** Botond Zsombor Pertics, Tamás Kovács, György Schneider

**Affiliations:** 1Department of Medical Microbiology and Immunology, Medical School, University of Pécs, Szigeti St. 12., H-7624 Pécs, Hungary; 2Department of Biotechnology, Nanophagetherapy Center, Enviroinvest Corporation, Kertváros St. 2., H-7632 Pécs, Hungary

**Keywords:** *Klebsiella pneumoniae*, Klebsiella phage, bacteriophage, capsule serotype, capsule depoly-merase, phage receptor, K2 serotype

## Abstract

*Klebsiella pneumoniae* is a nosocomial pathogen. Among its virulence factors is the capsule with a prominent role in defense and biofilm formation. Bacteriophages (phages) can evoke the lysis of bacterial cells. Due to the mode of action of their polysaccharide depolymerase enzymes, phages are typically specific for one bacterial strain and its capsule type. In this study, we characterized a bacteriophage against the capsule-defective mutant of the nosocomial *K. pneumoniae* 52145 strain, which lacks K2 capsule. The phage showed a relatively narrow host range but evoked lysis on a few strains with capsular serotypes K33, K21, and K24. Phylogenetic analysis showed that the newly isolated Klebsiella phage 731 belongs to the *Webervirus* genus in the *Drexlerviridae* family; it has a 31.084 MDa double-stranded, linear DNA with a length of 50,306 base pairs and a G + C content of 50.9%. Out of the 79 open reading frames (ORFs), we performed the identification of *orf22*, coding for a trimeric tail fiber protein with putative capsule depolymerase activity, along with the mapping of other putative depolymerases of phage 731 and homologous phages. Efficacy of a previously described recombinant K2 depolymerase (B1dep) was tested by co-spotting phage 731 on *K. pneumoniae* strains, and it was demonstrated that the B1dep-phage 731 combination allows the lysis of the wild type 52145 strain, originally resistant to the phage 731. With phage 731, we showed that B1dep is a promising candidate for use as a possible antimicrobial agent, as it renders the virulent strain defenseless against other phages. Phage 731 alone is also important due to its efficacy on *K. pneumoniae* strains possessing epidemiologically important serotypes.

## 1. Introduction

*Klebsiella pneumoniae* is an encapsulated, Gram-negative bacterium, omnipresent in the environment and also an opportunistic nosocomial pathogen [1,2]. Colonization of the human skin and mucosal surfaces (oropharynx and the gastrointestinal tract) is prevalent, presenting an origin of severe infections of the respiratory and urinary tracts, wounds, and catheter entry points; hospitalized and immune-compromised patients are typically susceptible to the progress of such infections to potentially life-threatening conditions and septicemia [1,3,4,5]. In the past few decades, community-acquired infections, e.g., metastatic meningitis, endophthalmitis, and pyogenic liver abscesses (PLA) [1] are also reported in young and healthy individuals. *K. pneumoniae* isolates are frequently resistant to multiple antibiotics; the pathogen is a member of the ESKAPE group of microorganisms (*Enterococcus faecium*, *Staphylococcus aureus*, *Klebsiella pneumoniae*, *Acinetobacter baumannii*, *Pseudomonas aeruginosa,* and *Enterobacter* spp.) [6].

The outermost layer on the bacterium, the capsule, acts as a physical barrier against host immunity and antibiotics [7]. It is involved mainly in resistance to phagocytosis, therefore it is considered as a crucial virulence factor. Capsular polysaccharides (CPS) have structural differences in the polysaccharide chains, and are classified into ≈80 serological types (K antigens) and into more than 140 genetically distinct capsular locus types [8]. Due to these differences (capsule thickness and glycan structure), the level of virulence is not equal between all serotypes [9]. On account of its resistance, the K2 is one of the most prominent serotypes, frequently collected from patients [5,10,11,12,13,14,15,16], with a particularly high prevalence in liver abscesses and endophthalmitis [17].

Application of bacteriophages (phages) is an emerging and promising solution for combatting antibiotic-resistant isolates. Phages are bacterial viruses that are able kill the target by recognizing specific receptor structures on the surface of the bacteria, attach to and infect the host cell, releasing phage progenies entailing the lysis of the cell itself [18].

Bacteria enveloped with polysaccharide capsules can be effectively controlled by phages: in vitro and in vivo applications against *K. pneumoniae* that have been accomplished [19,20,21,22,23,24,25,26,27,28,29,30]. Infection of capsulated *K. pneumoniae* strains require phages to get through the CPS by applying specific polysaccharide depolymerase enzymes, which recognize and degrade the CPS structure, allowing the phage itself to access the bacterial cell surface, adsorb to the outer membrane receptor, and infect the cell [31,32,33,34]. Capsule depolymerases have selectivity to certain serotypes (see Table 1 in [35] and Table 3 in this manuscript). Many phages exist that bear multiple depolymerases [36,37,38], targeting different capsular serotypes, empowering the phage to be multivalent, and have a broader host spectrum. From a therapeutical point of view, this could be a desirable trait. Another option is to apply bacteriophages recognizing conserved receptor structures and apply them together with small molecules which have the ability to shave off the capsule layer preventing phage binding.

In this study, we aimed to demonstrate this latter possibility by isolating a bacteriophage (731) targeting the capsule mutant of the hypervirulent *Klebsiella pneumoniae* 52145 [39] that is a thoroughly studied K2 reference strain [2,40,41], and apply together with the recently expressed K2 depolymerase (B1dep) [42].

Here we have demonstrated that the combined application of phage 731 together with B1dep depolymerase was able to lyse the hypervirulent K2 *K. pneumoniae* strain. Furthermore, phage 731 alone could lyse different *K. pneumoniae* strains possessing K21, K24, and K33 serotypes. This is the first detailed characterization of a phage able to lyse a K33 serotype.

## 2. Materials and Methods

### 2.1. Bacterial Strains and Growth Conditions

*Klebsiella pneumoniae* human isolate 52145 wild type (WT), its isogenic mutants, and 100 strains/isolates with diverse capsular serotypes, were used in this study (Table 1). Bacteria were grown on lysogeny broth agar (LBA) plates at 37 °C or liquid lysogeny broth (LB) medium (37 °C at 125 rpm). To produce a bacterial lawn, 100 μL of the overnight (ON) liquid cultures was plated onto a solid LB agar plate and incubated ON at 37 °C. Bacteria were proliferated ON in liquid medium at 37 °C in an orbital shaker (125 rpm).

### 2.2. Phage Isolation, Propagation, and Titer Determination

Klebsiella phage 731 was isolated in 2016 from a sewage farm (Pellérd, Hungary) with the traditional method [48]. Briefly, 1 mL of sewage sample was incubated with a 50 mL mid-log suspension (optical density OD_600_ = 0.5–0.6) of isolate 52145 capsule mutant (52145-Δ*wca*_K2_) ON at 37 °C. The suspension was centrifuged (4000 rpm, 10 min), chloroform was added to the supernatant at a 1:50 *v*/*v* ratio (Molar Chemicals Kft., Halásztelek, Hungary), and left ON at 4 °C. Spot testing [48] was used to detect the presence of lytic phages: those phages were selected, which formed lytic zones on the lawn of 52145-Δ*wca*_K2_, but not on the wild type (WT). Spot testing in this study was performed as follows: 10 µL phage suspension was dropped after plating 100 µL bacteria on LB agar. Plates were left to dry, then incubated ON at 37 °C. Individual phage plaques were excised using the agar overlay method [49] and were purified in 3 consecutive steps. The purified phage clone was named “Klebsiella phage 731” in accordance with the current phage nomenclature [50], was propagated in 100 mL LB medium, centrifuged (11,000 rpm, 30 min), and resuspended in 50 mL deionized water (DW). The phage titers were determined by spot testing of the serial dilution of the phage suspension, the plaque-forming unit (PFU) was calculated for 1 mL of the concentrated suspension. The resulting high-titer suspension of Klebsiella phage 731 (10^9^ PFU/mL) was used for further studies. High-titer suspensions of the phage were stored in 15 mL aliquots at 4 °C in LB and in 1.5 mL aliquots at −80 °C in LB:glycerol (80%) 2:1 *v*/*v.*

### 2.3. Determination of Host Range and Efficiency of Plating

Host range of phage 731 was determined by spot testing on 105 *K. pneumoniae* isolates possessing known capsule serotypes (Table 1). Spots were recorded and termed as (i) ‘clear’: phage can efficiently lyse the bacteria, full clearance is observable; (ii) ‘veiled’: partial lysis occurs with a turbid clearing zone; (iii) ‘no clearance’: the isolate is resistant to the phage, no effect is visible. Clear spots were observed solely without a turbid ring (halo), suggesting that depolymerase activity is not unambiguously detectable. Hence ‘veiled’ clearing forms were not assigned to be halo zones (capsule degradation only), as in the case of phage B1 [42].

*K. pneumoniae* isolates sensitive to phage 731 in the spot test were selected for determination of the efficiency of plating (EOP) as previously described [29], with some modifications. Briefly, the selected isolates were grown ON at 37 °C. After plating 100 µL, PFU of phage 731 was determined on the isolate. The EOP was defined as the average PFU on target bacteria/average PFU on host bacteria. Efficiency was classified as highly productive (EOP ≥ 0.5), moderately productive (0.1 ≤ EOP < 0.5), low productive (0.001 < EOP < 0.1), or inefficient (EOP ≤ 0.001). Results were reported as the mean of 3 independent measurements.

### 2.4. One-Step Phage Growth Curve and Adsorption Assay

One-step phage growth curve and burst size were determined as described previously [42], with modifications. Briefly, either the host strain (52145-Δ*wca*_K2_) or the CIP 53.8 was grown at 37 °C until log phase (OD_600_ = 0.5–0.6, 10^8^ CFU/mL), then 0.9 mL was mixed with 0.1 mL of the phage suspension (10^7^ PFU/mL) to achieve a multiplication of infection (MOI) of 0.01. The mixture was incubated for 10 min at 37 °C and subsequently centrifuged (13,000 rpm, 4 min). The pellet was washed with 1 mL LB to exclude any non-adsorbed phages from the medium. The pellet was resuspended in 1 mL LB, diluted 1:10,000 in 50 mL LB medium, and incubated at 37 °C with shaking. Aliquots of 500 µL were sampled from zero time to 1 h with 5 min intervals, and were treated with 1:50 *v*/*v* chloroform (Molar Chemicals Kft., Halásztelek, Hungary) following incubation at 4 °C ON. After centrifugation (12,000 rpm, 1 min), the PFU of the supernatant was determined by spot testing. The latency period was defined as the time between infection and the shortest incubation time, allowing the production of phages. The burst size was calculated as the ratio between the number of phage particles released at the plateau level and the initial number of infected bacterial cells. The experiments were performed 3 times, and the reported values are the mean of the observations.

Phage adsorption assay was performed as described previously [42]. Briefly, 1 mL of exponential-phase culture (10^8^ CFU/mL) was mixed with 10 µL of diluted phage (10^7^ PFU/mL) suspension (MOI = 0.01). The mixture was incubated (37 °C, 10 min) and centrifuged (13,000 rpm, 5 min). The titer of the supernatant was determined by spot test. The phage adsorption efficiency (*e*) was defined as $e=\frac{t-t_{s}}{t}$, where *t* is the initial phage titer and *t_s_* is the residual titer in the supernatant.

### 2.5. Transmission Electron Microscopy

Morphology of the phages was examined by transmission electron microscopy (TEM) as described recently [42]. Briefly, 10 µL from the purified high-titer (10^9^ PFU/mL) phage stock was deposited onto formvar-coated copper grids (Pelco Grids, Redding, Canada) and negatively stained with 1.5% *w*/*v* phospho-tungstic acid (Merck KGaA, Darmstadt, Germany) for 40 s. After drying, phages were visualized on a JEM-1400 Flash TEM (JEOL USA Inc., Peabody, MA, USA) operated at 80 kV acceleration voltage, with 54 µA beam current.

### 2.6. Phage DNA Extraction, Genome Sequence and Bioinformatic Analysis

DNA was extracted as described in our previous study [42]. Briefly, 1.5 mL phage suspensions (10^9^ PFU/mL) were centrifuged (10 min at 10,000 rpm). To the supernatant, 10 μL DNase I (1 mg/mL D4527, Sigma-Aldrich, dissolved in 0.15 M NaCl) and ≈10 mg RNase A crystals (R5503, Sigma-Aldrich, St. Louis, MI, USA) were added and incubated for 30 min at room temperature. After treatment with 100 μL 0.5 M EDTA (pH 8) (10 min, 75 °C), 20 μL Proteinase K (10 mg/mL, recombinant, PCR Grade, 03 115 852 001, Roche Diagnostics GmbH, Basel, Switzerland) was added (1 h, 65 °C). After supplementing with 50 μL 7.5 M ammonium acetate, phage DNA was extracted with phenol–chloroform (phenol/chloroform/isoamyl alcohol 25:24:1 *v*/*v*/*v*, saturated with 10 mM Tris (pH 8.0) and 1 mM EDTA, P2069, Sigma-Aldrich, St. Louis, MI, USA) and precipitated with 96% ethanol.

The purified phage DNA was dissolved in 100 μL of sterile nuclease-free H_2_O and was used to prepare genomic DNA sequencing libraries by using the Nextera XT Library Preparation kit (Illumina, San Diego, CA, USA). Sequencing was performed using the MiSeq Reagent Kit v2 (2 × 150 bp) on an Illumina MiSeq instrument (Illumina, San Diego, CA, USA). The Mypro pipeline was used to assemble the gained pure sequences.

The assembled sequence was annotated on the RAST server (https://rast.nmpdr.org/, access date: 22 December 2022), CLC Sequence Viewer v.6 (CLC bio, Aarhus, Denmark) was used to analyze, and Easyfig 2.2.5 (https://mjsull.github.io/Easyfig/, access date: 22 December 2022) to illustrate the genome maps. Open reading frames (ORFs) and gene predictions were verified by GeneMarkS [51]; genome was searched for restriction endonuclease recognition sites in silico by Webcutter 2.0 online (http://heimanlab.com/cut2.html, access date: 22 December 2022). Homology searches were conducted by the BLAST tools available at the NCBI website (https://blast.ncbi.nlm.nih.gov/Blast.cgi, access date: 19 March 2021). Phage was classified according to the guidelines of the International Committee on Taxonomy of Viruses (ICTV, talk.ictvonline.org/taxonomy/, access date: 22 December 2022) confirmed by ViralZone (viralzone.expasy.org access date: 22 December 2022) and BLASTn results. Protein homology, conserved domain prediction and protein characterization was conducted by NCBI BLASTp, InterProScan (http://www.ebi.ac.uk/Tools/pfa/iprscan/, access date: 22 December 2022), NCBI COBALT (Constraint-based Multiple Alignment Tool, https://www.ncbi.nlm.nih.gov/tools/cobalt/re_cobalt.cgi, access date: 22 December 2022), Jalview 2.11.2.5 (https://www.jalview.org/, access date: 22 December 2022), and Protparam tool (https://web.expasy.org/cgi-bin/protparam/protparam, access date: 22 December 2022) visualization of the putative depolymerase was performed by SWISS-MODEL (https://swissmodel.expasy.org/interactive, access date: 22 December 2022).

The nucleotide sequence of phage 731 was deposited in the GenBank database under the accession number OQ404738.

### 2.7. Phylogenetic Analysis of Phage 731

Whole genome-based phylogenetic analysis was performed with VICTOR [52], involving the first 32 highly similar *Klebsiella* phages, according to the homology searches and our *Klebsiella* phages.

All pairwise comparisons of the nucleotide sequences were conducted using the Genome-BLAST Distance Phylogeny (GBDP) method [53] under settings recommended for prokaryotic viruses [52]. The resulting intergenomic distances were used to infer a balanced minimum evolution tree with branch support via FASTME including SPR postprocessing [54]. Branch support was inferred from 100 pseudo-bootstrap replicates each. Trees were rooted at the midpoint [55] and visualized with ggtree [56]. Taxon boundaries at the species, genus, and family levels were estimated with the OPTSIL program [57], the recommended clustering thresholds [52], and an F value of 0.5 [58].

## 3. Results

### 3.1. Morphological Features of Phage 731

From a pool of 20 selected and amplified plaques purified from wastewater with strong lytic characteristics against the capsule mutant (K-) 52145-Δ*wca*_K2_ *K. pneumoniae* strain, four, that were additionally unable to cause lysis on the 52145 WT strain, were subsequently isolated and further purified. Similar restriction patterns with *Eco*RI and *Hind*III were considered when we chose one phage (731) for evaluation.

TEM analysis showed that phage 731 has a ≈50–60 nm head and a 150–200 nm long non-contractile flexible tail, which is a feature of the former *Siphoviridae* family of the tailed phages (Figure 1d).

The phage formed 1.5–2.5 mm diameter individual plaques on its own host 52145-Δ*wca*_K2_. Around the clear phage 731 plaques, no turbid rings (halo zones) were observed after ON incubation (Figure 1a). The phage formed clear but smaller (0.2–0.5 mm) plaques on the 53.8 (K33) strain, also without halos (Figure 1b). No halos were observed on either of the lawns after months of storage.

### 3.2. Growth Characteristics of Phage 731

One-step growth experiment was conducted to determine the burst size of phage 731 on its host 52145-Δ*wca*_K2_ (K-) and on strain 53.8 (K33). On the host, a triphasic curve was obtained with a latent, a log/rise, and a plateau period (Figure 1c). Phage 731 had a 10-min latency, and the plateau level was reached at 30 min. Burst size was ≈1000 phage particles per infected bacteria. On strain 53.8, latent period was between 5–10 min, the plateau was reached at 25 min and the burst size was ≈2000. Adsorption efficiency of phage 731 on its own host was 99.54% and 99.6% on strain 53.8 (K33). Adsorption was tested on strains ABC606 (K21), ABC360 (K24), and 52145-Δ*wca*_K2_Δ*waaL* (O-:K-), and was 99.3%, 99.95%, and 97%, respectively.

### 3.3. Host Range and Efficiency of Plating

Spot tests and PFU determinations were performed on a *K. pneumoniae* isolate collection, containing 105 isolates with 37 different capsular serotypes and 3 additional CPS/LPS mutants, to measure the host range of phage 731, and expand the previously published spectrum of phage B1 and B1dep [42].

Results showed that even though phage 731 was isolated against the 52145-Δ*wca*_K2_ capsule-less mutant, it showed a different level of effectivity altogether for 8 capsular serotypes from our collection. It formed clear, halo-less plaques and exhibited high efficiency (EOP = 1) on strains ABC606, ABC360, and CIP 53.8, with capsules K21, K24, and K33, respectively. However, on other K21 and K24 isolates, there was no effect visible (except for one K21, see below). Efficiency was also high on the double mutant 52145-Δ*wca*_K2_Δ*waaL* (O-:K-).

Phage 731 was also showing some minor lytic activity against the following strains: CIP 52.215, CIP 52.224, CIP 52.225, CIP 52.232, PFZ334, PFZ155, and CIP 80.47, having K11, K20, K21, K27, K27, K51, and K64 serotypes, respectively. Spot testing resulted a ’veiled’ clearing, which was not growing upon time, unlike halo zones. Spotting diluted suspensions resulted tiny, indiscernible individual plaques on the lawn, within the dropping area. This most likely means incomplete lysis, and the phage was considered ineffective against these strains.

Combination of phage 731 with the recombinant B1dep protein resulted in a significant effect when dropped together on the wild-type 52145 strain (Figure 2). As the B1dep degraded the K2 capsule and produced halo spots, phage 731 became able to evoke clear lysis on the lawn, within the halos. Nevertheless, the phage was not effective on other strains, which capsule. Was susceptible to the B1dep alone.

However, supportive effect of B1dep was demonstrated with another phage from the *Webervirus* genus, phage 13 (see Section 3.4), which proved to be specific for and effective against 40 different isolates with K24 capsule [29]. It also performed full, clear lysis on the 52145 strain, but only in combination with B1dep (Figure 2). It suggested that phage 13 engages the same membrane receptor as 731 and/or B1 on 52145, even though it is not able to get through the K2 capsule.

### 3.4. Genomic and Phylogenetic Properties of Phage 731

Basic genome statistics revealed that phage 731 has a 31.084 Mda double-stranded, linear DNA with a length of 50,306 base pairs and a G + C content of 50.9% (25.7% adenine, 24.6% cytosine, 26.4% guanine, and 23.4% thymine). The genome encompasses 79 ORFs, 60 on the positive, 19 on the negative strand. The average gene length is 581 bp. Remnants of mobile genetic elements and traces of truncated genes were not observed.

Closest databank homologies of Klebsiella phage 731 are listed in Appendix A. The first 4 hits are Klebsiella phage vB_KpnD_PeteCarol (OL539448.1), GML-KpCol1 (NC_047907.1), PhiKpNIH-10 (MN395285.1), and MezzoGao (NC_047850.1). Besides these, phage 731 showed high similarity (coverage > 80%, identity > 90%) with more than 100 Klebsiella phages, such as 1513 (KP658157.1), KLPN1 (KR262148.1), PKP126 (KR269719.1), and the above mentioned Klebsiella phage 13 (coverage = 76%, identity = 96.01%, NC_049844.1); all are members of the *Webervirus* genus (*Drexlerviridae* < *Caudoviricetes* < *Uroviricota* < *Heunggongvirae* < *Duplodnaviria*, according to the recent ICTV taxonomy release, March 2022).

Phylogenomic analysis of phage 731 with other Klebsiella phages reveals the closest relation to Klebsiella phage B1 (MW672037.1) and KPN N141 (NC_047841.1) (Figure 3).

The comparison of the genome annotations of the simultaneously isolated phages B1 and 731 was performed (Figure 4). Coverage was 90% with a 96% sequence identity. The comparison reveals that the overall high similarity is impaired only at a definable gene, *orf61* of B1 phage, which is a K2 depolymerase and was cloned and expressed as B1dep in our previous study (Figure 5, striped arrow). The corresponding segment of the phage 731 genome is the 2918 bp long *orf22*, starting at roughly 17.9 Kb and ending at 20.8 Kb and is highly dissimilar with *orf61* of B1.

As the only difference between bacterial hosts 52145 WT and 52145-Δ*wca*_K2_ (K-) is the presence of the capsule, the only presumable distinction between the two phages is the presence and/or specificity of the capsule depolymerase. The selected *orf22* region was assumed as a putative depolymerase and was further investigated.

### 3.5. Molecular Properties of the Putative Phage Depolymerase

The putative depolymerase coding gene of phage 731 (Figure 5), *Orf22* is 2918 bp long and codes for a 102.9 kDa, 972 aa protein, annotated as ‘Phage tail fiber protein’. It showed strong similarity with tail fiber proteins of *Klebsiella* phage GML-KpCol1 (95.37%, YP_009796909.1), vB_KpnS-VAC7 (92.6%, QZE50811.1), vB_KpnS_15-38_KLPPOU149 (92.49%, YP_009903158.1), and others, many with coverage over 90% (Appendix A). A conserved peptidase_S74 domain was reported by NCBI BLASTp between residues 840–962, which was also predicted by InterProScan. This domain proved to be highly conservative and showed no differences between the phages, according to the alignments performed with NCBI COBALT, Jalview, and Easyfig.

To detect whether the capsule specificity really only depends on this gene, the genome of phage 731 was also compared to different Klebsiella phages from the same genus, which were selected according to the NCBI BLASTn and BLASTp searches. This time, the genomes were truncated and the examination was focused on the genes around the *orf22*, including mostly tail regions.

The comparison was performed in 3 blocks. In the first block, phage 731 was aligned to phages, highly similar to each other, known to be specific for the K3 capsule: SegesCirculi, Domnhall, and KingDDD (Figure 6a). The rest of the phages were divided between the second and third blocks (Figure 6b,c), and were selected into the comparison according to the nearly 100% coverage of their gene corresponding to *orf22*. The goal of the assortment of phages and placing one phage next to another was to aim the best visualization array for genome homologies and differences. Capsule specificities of the phages are listed in Table 2.

It is clearly observable that there are only minor differences between the phage regions. Gene 22a-b hypothetical/phage proteins are not homologous between the phages, hence these regions may be responsible for host capsule recognition. Also, the variable regions of gene 23 (*orf22* of phage 731) phage tail fiber proteins (Figure 5a) near the N-terminal and the central region should be highlighted for the same reason. Another putative depolymerases have been identified in these genomes. In phages Domnhall, KingDDD, and SegesCirculi, there is a putative depolymerase (Figure 6a, gene 12), which is not present in the rest of the phage genomes. This suggests that this depolymerase may be responsible for K3 capsule specificity, as (according to the recent literature) the other phages are not specific for this serotype.

## 4. Discussion

In this study, we presented the isolation and characterization of Klebsiella phage 731. Isolation of this phage was necessary and had 3 purposes: (i) to support the characterization, depolymerase identification, and host/receptor specificity charting of phage B1, (ii) to test the K2-eradicating ability of the B1dep protein by applying together with phage 731, and (iii) to check if B1 resistant colonies are susceptible to phage 731 by losing their capsule upon resistance acquisition.

Identification of depolymerases had become less ponderous up until the present. Publications of characterizing Klebsiella phage capsule depolymerases, along some guidelines and with routine methods, have started to ascend in the last couple of years. Around 2016, one could find more or less only predictions of putative depolymerases in publications, very few gene bank data about capsule depolymerase proteins, and only a couple of studies, cloning and well-characterizing these enzymes [67,68,69]. However, the number of publications describing Klebsiella phage depolymerases increased until 2022: several were identified and characterized, acting specifically on different capsule serotypes (Table 3). Squeglia et al. in 2020 [35] and were the first to provide a precise, detailed structural and functional characterization of a Klebsiella phage depolymerase–capsule interaction of KP32gp38 tailspike and K21 capsule, and Dunstan characterized a depolymerase of a K2 strain [70].

Focusing on the phage–host relations with the 52145 variants, phage 731 is ineffective against the 52145 WT strain, as it is not able to degrade K2 capsule, but probably recognizes a membrane receptor beneath, masked by the capsule (Figure 7). Phage receptors are typically outer membrane proteins (OMP), capsule polysaccharides (CPS), and lipopolysaccharides (LPS) [71]. If a phage is not able to recognize and digest the reversible (secondary) receptor CPS or LPS, it is unable to get access to the irreversible (primary) receptor on the cell surface [34,62,72]. Irreversible receptors are typically OMPs. For GH-K3 phage, outer membrane protein C (OmpC) is essential for phage infection of a K2-encapsulated bacterial strain, while NJS1 phage employs FepA as an irreversible receptor, and OmpC loss has no significant effect on infection [62,73].

Lytic activity of phage 731 on double mutant 52145-Δ*wca*_K2_Δ*waaL* (O-:K-) suggests that the phage needs neither capsule nor O antigen to bind, and the primary receptor is on the bacterial surface, being probably an outer membrane protein. Lysis of phage B1 on the double mutant indicates that the capsule is not the exclusive receptor of this phage, and thus, the depolymerase activity is also not necessary for lysis, even though this model was suggested in different studies [74,75]. K2-type capsule stated both necessary and sufficient for RAD2 phage infection, CPS was identified as an essential receptor [70]. It was confirmed by the resistance of an isogenic acapsular mutant of the same strain to the RAD2 phage, just as 52145-Δ*wca*_K2_ resistance to phage B1 [42].

Also, presence of CPS is not necessary, and only has a role in phage recruitment by enhancing adsorption [67,73]. This is consonant with the less low EOP of phage B1 on the double mutant 52145-Δ*wca*_K2_Δ*waaL* compared to the WT strain.

Even so, why the B1 phage was unable to lyse the capsule-less mutant is most likely not the absence of a capsule as a receptor, but because the long chain length of an O antigen could be a barrier for phage infection (Figure 7) [62]. This implies that a shorter O-antigen chain may render bacteria more sensitive to phage infection, most likely by letting phages to attach to irreversible receptors more efficiently, as in the case of the double mutant 52145-Δ*wca*_K2_Δ*waaL* [62]. Also, LPS is an irreversible receptor only in the manner that it keeps the structure of the reversible receptor, CPS. In *K. pneumoniae* 52145, the LPS core has a direct role in the retention of capsule [76,77].

Phage 731 was isolated against 52145-Δ*wca*_K2_ [45], which is the capsule-less mutant of 52145. Isogenic capsule mutant strains are commonly avirulent, thus are not capable of causing pneumonia or UTI [2,78]. In our previous study, we already showed that B1-resistant mutant colonies are losing their capsule and become sensitive to phage 731. This phenomenon was also described elsewhere [61,79]. Cai and colleagues presented that GH-K3 phage-resistance mutants have their CPS coding genes down-regulated, whether OmpC expression levels are unchanged, thus resistance is independent of the irreversible receptor. The genome of the bacteria did not change, only the protein expression levels, and acapsular mutants did not gain back their capsule. CPS mutants also had more amount of LPS, which masked membrane receptors [80]. This requests the potential use of phages B1 and 731 in a cocktail.

Expanding the scope to the other listed *K. pneumoniae* strains (Table 1), since our previous study [42], it was possible to test the phages on the rest of the strains as they have become available. The host spectrum of Klebsiella phage B1 is narrow, and it is specific only for strain 52145, in terms of full lysis. The recombinant B1dep depolymerase, however, according to our experiments, is not exclusively K2 specific, as it formed halo zones on one K24 and on one K106 strain. The phenotype of these strains needs further examination, to assure whether the B1dep is K2 specific, or also act on other capsules. If this latter is the case, B1dep would be the first described polyvalent Klebsiella phage depolymerase to our knowledge. The equivalent spectrum of B1 phage and B1dep halo zones suggested that B1 has only one depolymerase, and if there is polyvalency, then it is the B1dep, not the phage. It seems that the specificity of the B1 phage itself is limited by its depolymerase, and through this, it also has a narrower spectrum than the enzyme.

By contrast, phage 731 unexpectedly proved to have a broader host spectrum.

Due to the clear lysis on strains with capsule serotype K21, K24, and K33 and veiled lysis on 6 further serotype, phage 731 alone may also be a potential therapeutic agent.

K21 is a predominant serotype [81,82,83], while K24 was reported to be associated with sequence type-15 (ST15) [84,85,86] and frequently reported from other carbapenemase-producing *Klebsiella* strains [8]. K33 is not a markedly significant capsular serotype in terms of virulence or epidemiological prevalence, however, it was reported and/or examined in a set of studies in Europe and Asia [87,88], most of them investigating multidrug-resistant (MDR) *K. pneumoniae* isolates [89,90,91,92,93]. K33 was reported prevalent in urinary tract infections (UTIs) [94], isolated as nosocomial pathogen from urine [90], blood [88], from ventilators and bedrails of an ICU [95], and also reported as the most common serotype along with K69 in surface water isolates [96]. Strains with K33 serotype were also isolated from unrelated pediatric patients in China (neonatal pneumonia, from feces) [97] and in Mexico (sepsis, from blood) [98]. K33 is also a strong biofilm producer and resistant to antihistaminic drugs [99] and shows cross-reactivity with K35 serotype [100,101].

To date, only a few, non-characterized phages are published against K33 capsule [100,102]. Pieroni et al. describe ϕ33 and ϕ35 isolated against K33 and K35, also active on each other’s host with lower efficiency. Plaques of ϕ35 are described clear and halo-less, just as the plaques of phage 731 on the K33 lawn. Portilla Rincón described 4 phages (F1, F2, F6, F7), which were active against a carbapenemase-producing K33 strain. Our phage 731 is the first well-characterized phage against the K33 capsule.

Polyvalence of particular phages are due to that they are equipped with more depolymerases, and their broad lytic spectrum is matching with the sum of specificity range of their individual depolymerases [38,74,103]. To date, every described Klebsiella phage depolymerases were specific to one capsular serotype (Table 3). This suggests that phage 731 has more than one depolymerase rather than one specific for more capsule serotypes.

Specificity of phages and their depolymerases can even be more delicate than the capsule serotype level, due to the subtle differences in the sugar chain composition [104]. Some polyvalent phages are selective to different strains with the same capsule type [36], just as in the case of 731 (e.g., K21 or K24 strains).

We wanted to unravel the reason behind the polyvalence. Comparing with homologous phages, there was no major difference in the genomes regarding the related ORFs. The only discrepancy was the gene, numbered 12 on Figure 6. is probably responsible for K3 specificity, as it is present only in the three, closely related phages, specific for the K3 capsule. While the peptidase region is conserved in gene 23 (*orf22* in phage 731), divergence is observable near the N-terminal, and also between hypothetical proteins 22a and 22b—these alterations are present nearly between all phages compared in this study. As these 731-related phages are not really described in the literature regarding depolymerase activity and capsule specificity, the topic requires further research, including the cloning and expression of individual ORFs. The case is more subtle if we consider that phage 731 showed no observable halo zones on none of the host lawns even after months, which could mean that its depolymerase is non-soluble and part of a tail fiber or tail tip [105,106]. Examination of these phages is worth considering, as many of them were isolated against capsule-less mutants of bacterial strains, and proved to be more or less effective on different capsules [61,62]. This trait is not exclusive and was published for many Klebsiella phages [36,38]. It is also possible that the membrane receptor somehow extrudes from these capsules, thus the capsule cannot act as a barrier and does not affect phage infectivity.

As it was mentioned, one of the purposes of Klebsiella phage 731 was to test the K2 capsule degrading ability of the previously expressed B1dep depolymerase in a functional analysis.

Successful lysis after spotting the mixture of B1dep and phage 731 or 13 proved on one hand, that B1dep deprives K2 encapsulated bacteria from their CPS, thus degrades the significant virulence factor, the capsule, enabling other phages (which are originally not active on 52145 WT) to lyse the bacterial cells. B1dep-731 mixture ineffectiveness on the rest of our bacterial strains may be due to that these strains employ a different irreversible receptor as 52145 that phages 731 and 13 cannot recognize. Testing other phages in combination with B1dep is scheduled in the future. On the other hand, the activity of phage 13 and 731 on 52145 supported the fact that, by capsule degradation, phages embarking depolymerases that act on different capsule serotypes may gain higher lytic spectrum by reaching previously masked irreversible outer membrane receptors, enabled by independent, recombinant depolymerases [107]. This also suggests that irreversible receptor specificity of particular phages is independent of capsule specificity and capsule depolymerase arsenal.

**Table 3 microorganisms-11-00669-t003:** List of hitherto published Klebsiella phage depolymerases. List is focusing on studies (in chronological order), where a depolymerase was expressed or at least a putative depolymerase was mentioned. Putative depolymerase names are in *italic*, protein names are in **bold**, if depolymerase was expressed. Unknown capsular serotypes were exchanged by the strain code. Asterisks (*, **) are indicating the same ORF/depolymerase from the same authors. ND = not defined. Question mark means uncertainty in the related information.

Klebsiella Phage	DepolymeraseGene/Protein	Capsule Type	Author, Year of Publication	Reference
0507-KN2-1	**ORF96**	KN2	Hsu et al., 2013	[67]
NTUH-K2044-K1-1	**K1-ORF34**	K1	Lin et al., 2014	[68]
KLPN1	*ORF43*? *ORF34* and/or *ORF35*	K2	Hoyles et al., 2015	[108]
P13	*gene 49*, *gene 50*	K13	Shang et al., 2015	[109]
K5	*ORF40*, *ORF41*	(*K. pneumoniae* KSM 5-1)	Shneider et al., 2016	unpub-lished
KP36	*gp50* **depoKP36**	K63	Majkowska-Skrobek et al., 2016	[69]
KpV289	*kpv289_orf45*		Volozhantsev et al., 2016	[110]
K5-2 K5-4	*ORF37* **K30/K69dep** *ORF37* **K8dep** *ORF38* **K5dep**	K30, K69 K8 K5	Hsieh et al., 2017	[74]
ΦK64-1	**S1-1** **S1-2** **S1-3** **S2-1** **S2-2** **S2-3** **S2-4** **S2-5** **S2-6** *S2-7* *S2-8*	K11 KN4 K21 KN5 K25 K35 K1 K64 K30, K69 / /	Pan et al., 2017	[38]
KP32	**KP32gp37** **KP32gp38**	K3 K21	Majkowska-Skrobek et al., 2018	[69]
KpV41 KpV475 KpV71 KpV74 KpV763 KpV767 KpV766 KpV48	*kpv41_55, kpv41_46* *kpv475_51* *kpv71_52* **Dep_kpv71** *kpv74_56* **Dep_kpv74 **** *kpv763_43* *kpv_767_46** *kpv766_44* *kpv48_57*	K1 K1 K1 K2 K2 K57 ND ND	Solovieva et al., 2018	[111]
SH-KP152226	*ORF42* **Dep42**	K47	Wu et al., 2019	[112]
vB_KpnP_IME321	*ORF42* **Dp42**	KN1	Wang et al., 2019	[113]
vB_KpnS_GH-K3	*gp32*	K2	Cai et al., 2019	[114]
KN1-1 KN3-1 KN4-1	**KN1dep** **KN3dep, K56dep** **KN4dep**	KN1 KN3, K56 KN4	Pan et al., 2019	[75]
13	*ORF2*	K24	Horváth et al., 2020	[29]
KpV79 KpV767	*kpv79_42* **Dep_kpv79** *kpv_767_46* **Dep_kpv767 *****	K57 K57	Volozhantsev et al., 2020	[115]
πVLC5 πVLC6	*ORF49, ORF58* *ORF51, ORF58*	K22, K37 K22, K37, K13 (halo: K2, K3)	Domingo-Calap et al., 2020	[28]
IME205	*ORF42* **Dpo42** *ORF43* **Dpo43**	K47 K47 *(different bac. strains)*	Liu et al., 2020	[95]
SH-KP152410	*ORF41* **K64-ORF41**	K64	Li et al., 2021	[116]
B1	*orf61* **B1dep**	K2	Pertics et al., 2021	[42]
GBH001 GBH014 GBH038 GBH019	*GBH001_048* **GBH001_056** *GBH014_001* *GBH014_051* **GBH038_054** **GBH019_279**	K1 *(not active)* K1 K2 *(not active)* K2 *(not active)* K2 K51	Blundell-Hunter et al., 2021	[117]
KpS8 vB_KpnP_Dlv622 vB_KpnM_Seu621	*kps8_053* **DepS8** *dlv622_00059* **Dep622** *Seu621_orf00052*	K23 K23 K23	Gorodnichev et al., 2021	[118]
SRD2021	*ORF58*	K47	Hao et al., 2021	[119]
RAD2	*gp02* **DpK2**	K2	Dunstan et al., 2021	[70]
KpnM6E1	**gp86**		Nogueira et al., 2021	[120]
P24 P39	*gene 11* *gene 81*	K64 K64	Fang et al., 2022. jan.	[121]
P13	*gene 22, gene 23*	K47	Fang et al., 2022. febr.	[122]
P560	*ORF43* P560dep	K47	Li et al., 2022	[123]
1611E-K2-1	*ORF16* K2-ORF16	K2	Lin et al., 2022	[124]
vB_KpnM-VAC13 vB_KpnM-VAC66	*ORF104* *ORF79*	K13, K15, K24, K36, K52, K64, K102, K107	Pacios et al., 2021, 2022	[37]
P929	*ORF56*	K19	Chen et al., 2022	[125]
KPR2	*CDS 48* *CDS29*	(*K. pn.* 033)	Reales-González et al., 2022	unpublished
SH-KP156570	*ORF41* **K19-Dpo41**	K19	Hua et al., 2022	[126]
vB_kpnM_17-11	*orf022* **Dep022**	K19	Bai et al., 2022	[127]
KpV74	*kpv74_56* **Dep_kpv74 ****	K2	Volozhantsev et al., 2022	[128]
731	*orf22*	K33, K24, K21 (?)	this study	

## 5. Conclusions

Our results propose that phage 731 is a promising contender for postliminary research. Although this phage was originally isolated against an avirulent mutant of a hypervirulent serotype, it showed effectivity on different serotypes, and proved to be polyvalent. All its examined properties were suitable, and it is worth further characterization. Our results might help the improvement of therapeutic approaches for *Klebsiella pneumoniae* infections, targeting different nosocomial strains, and also extend our knowledge about bacteriophage depolymerase specificity relations and mechanisms of such phages.

## Figures and Tables

**Figure 1 microorganisms-11-00669-f001:**
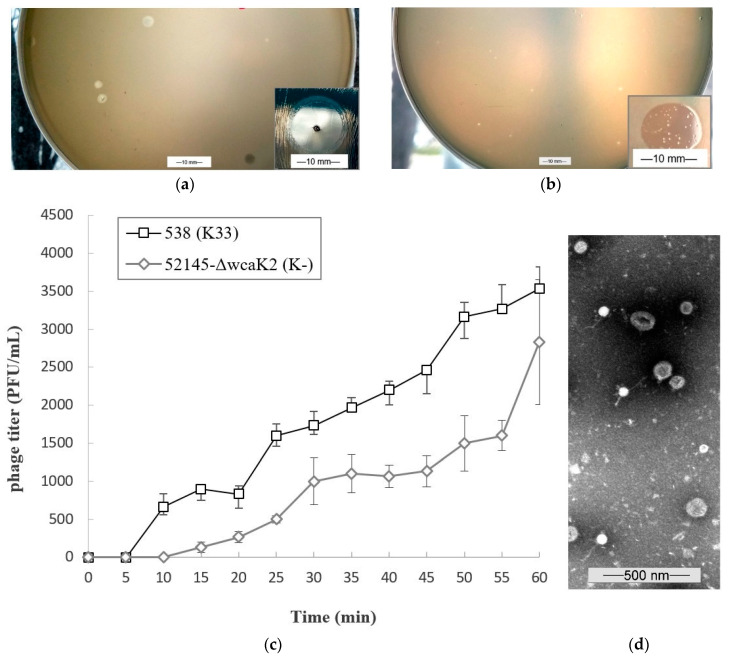
Characteristics of phage 731. (**a**,**b**) Individual plaques and clear spot (small pictures) on 52145-Δ*wca*_K2_ (K-) lawn (**a**) and on 53.8 (K33) lawn (**b**). Individual plaques were acquired by agar overlay method, clear spots were gained by dropping 10 µL of the concentrated phage suspension. Scale bars represent 10 mm. (**c**) One-step growth curves of phage 731 on the 52145-Δ*wca*_K2_ (K-) (gray) and on the 53.8 (K33) strain (black). The titer of the phage at different times are shown. Data are the mean of 3-3 independent experiments, error bars represent ± SD. (**d**) Electron micrograph of phage 731 stained with 1.5 *w*/*v* hosphor-tungstic acid. Scale bar represents 500 nm.

**Figure 2 microorganisms-11-00669-f002:**
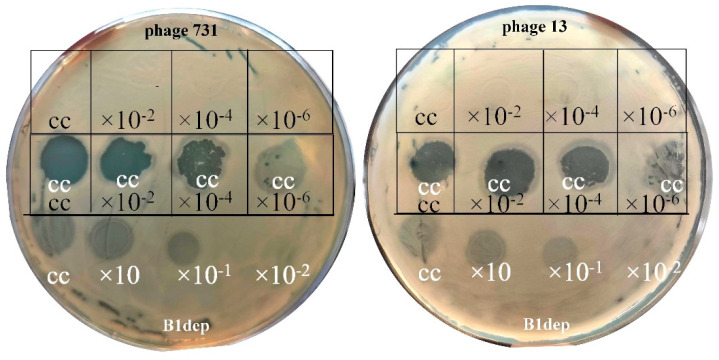
Synergistic effect of B1dep protein and phages 731 and 13 on 52145 lawn with the K2 capsule. Middle rows: 10 microliters of the purified recombinant B1dep protein was mixed and spotted along with 10-10 µL-s of different titers of phage 731 (**left** plate) or phage 13 (**right** plate) suspensions. Phage control dilutions (**up**) and B1dep control with dilution (**bottom**) were also spotted and are presented. Clear zones (dark here) indicate full lysis of the bacteria by the phages; semi-translucent (light dark here) halo zones are produced by B1dep. Cc = concentrated phage suspension (≈10^9^ PFU/mL, black text) or B1dep (≈800 ng, white text).

**Figure 3 microorganisms-11-00669-f003:**
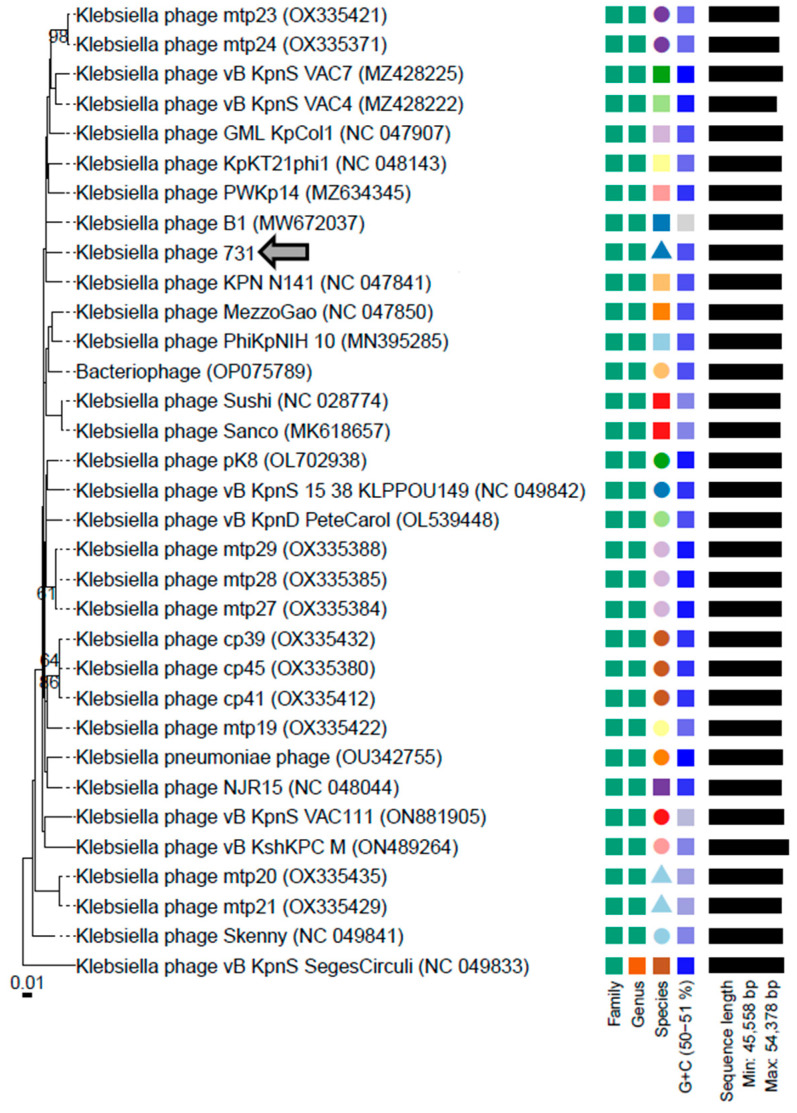
Whole genome-based phylogenetic relations of phage 731. Genome–BLAST distance phylogeny (GBDP) trees were inferred using the formula D4 and yielded average support of 17%. The numbers above branches are GBDP pseudo-bootstrap support values from 100 replications. The branch lengths of the resulting VICTOR trees are scaled in terms of the respective distance formula used. The OPTSIL clustering yielded 26 species clusters, 2 clusters at the genus level, and 1 at the family level. Accession numbers are also indicated next to the phage names. Klebsiella phage 731 is marked with a grey arrow (
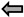
).

**Figure 4 microorganisms-11-00669-f004:**
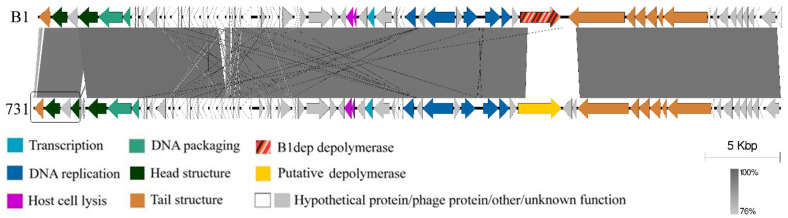
Comparative genome analysis of the previously isolated Klebsiella phage B1 and phage 731. Annotations of the genomes were done by the RAST server and were visualized by Easyfig. Predicted gene function groups are indicated.

**Figure 5 microorganisms-11-00669-f005:**
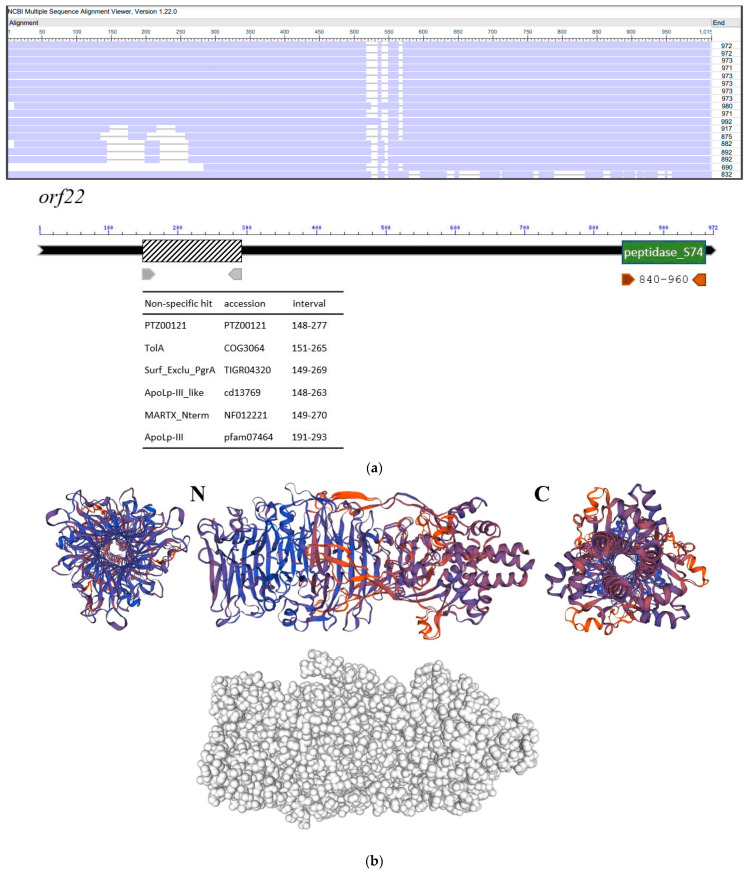
Molecular properties of the tail fiber protein coded by *orf22* of phage 731. (**a**) Comparison of homologous proteins of related phages by NCBI COBALT, the order of the phage proteins was originally aligned by Jalview and is the following from the top: 731, GML-KpCol1, vB_KpnS-VAC4, vB_KpnS-VAC7, vB_Kpn-VAC111, vB_KpnS_15-38_KLPPOU149, pK8, MezzoGao, vB_KpnD_PeteCarol, PhiKpNIH-10, NJS2, NJS1, Sweeny, vB_KpnS_Domnhall, vB_KpnS_KingDDD, vB_KpnS_SegesCirculi, PWKp14, Sushi. Pages are detailed in Table 2. Conserved regions of *orf22*-coded tail fiber protein of phage 731 are shown below, generated by NCBI BLASTp. (**b**) 3D model of *orf22*-coded tail fiber protein by SWISS-MODEL: ribbon model of the trimerized protein viewed from the N-terminal (**left**), side (**middle**), C-terminal (**right**), and space-filling model (**bottom**).

**Figure 6 microorganisms-11-00669-f006:**
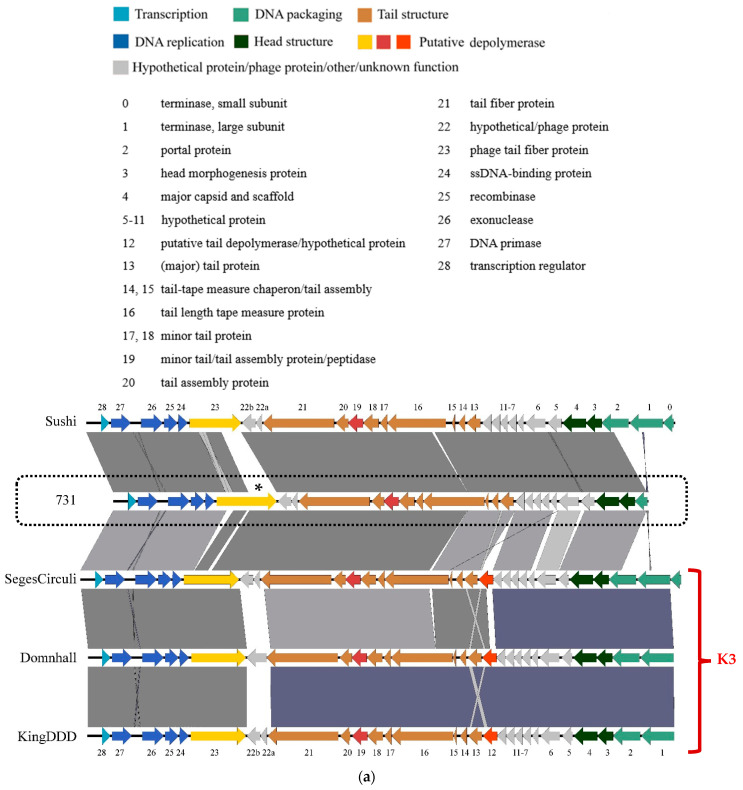
Genome similarities and differences of phage 731 to the representatives of the *K. pneumoniae* phages of the *Webervirus* genus. (**a**) Comparison of phages Sushi, 731, vB_KpnS_SegesCirculi, vB_KpnS_Domnhall and vB_KpnS_KingDDD. (**b**) Comparison of phages NJS2, NJS1, 731, PhiKpNIH-10, MezzoGao and vB_KpnD_PeteCarol. (**c**) Comparison of phages GML-KpCol1, 731, Sweeny, vB_KpnS-VAC4, vB_KpnS-VAC7, vB_Kpn-VAC111, vB_KpnS_15-38_KLPPOU149, pK8 and PWKp14. Genomes of the phages are truncated to the tail regions, containing putative depolymerases. Phage names are indicated on the left. Capsule specificities, where known, are written on the right. Phage 731 is framed and *orf22* is marked with an asterisk (*). The provided ORF numbering was used only in this study for practical reasons and is indicated on (**a**), predicted ORF functions are grouped, and their color codes are also indicated. Annotations of the genomes were performed by the RAST server and were visualized by Easyfig. Phages are listed in Table 2.

**Figure 7 microorganisms-11-00669-f007:**
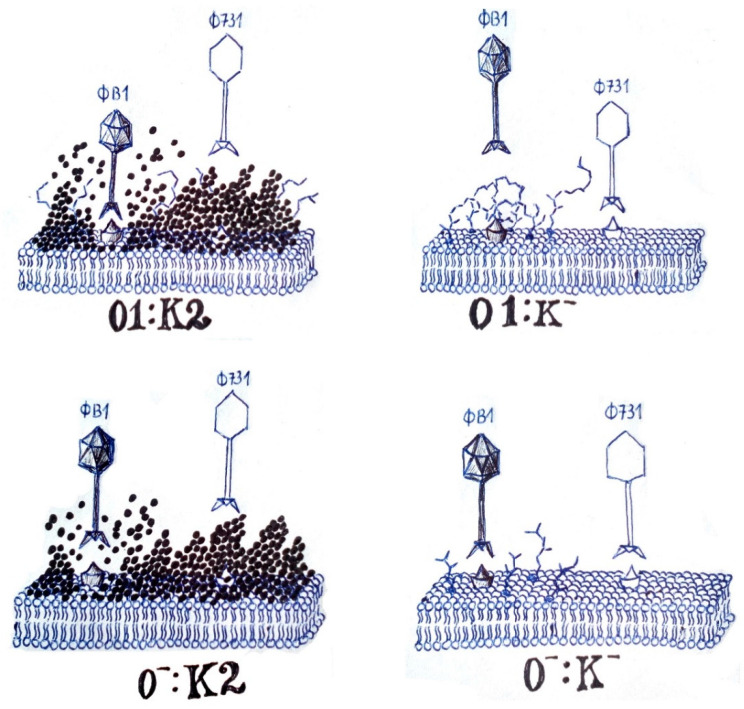
Hypothetical representation of the effect of phage B1 and 731 on *K. pneumoniae* 52145 wild-type and isogenic mutants. Upper left: wild type (52145); bottom left: LPS mutant (52145-Δ*waaL*); upper right: CPS mutant (52145-Δ*wca*_K2_); bottom right: LPS-CPS double mutant (52145-Δ*wca*_K2_Δ*waaL*). Phage efficiency is represented by the distance of the phage to the membrane. Outer membrane is represented as cross-segment, peaky spots are OMPs (irreversible receptors), LPS and long O antigen chains are represented by zig-zags, CPS is represented by the black dots.

**Table 1 microorganisms-11-00669-t001:** Bacterial strains used in this study and host range of phages 731, B1, and B1dep protein. Results were obtained by spot testing and are the mean of three different experiments. +++: clear lysis; +: veiled lysis; -: no effect; *H*: turbid ring around lysis (halo); H: turbid spot without lysis; h: less translucent turbid spot without lysis, smaller in diameter; *: B1dep was additionally tested on the grown (1-day old) lawn of the given strain; ND: no data available; question mark means uncertainty in the determined serotype. EOP classes are indicated where they were determined as ‘high’ (EOP ≥ 0.5) or ‘low’ (0.001 < EOP < 0.1).

	Strain Code	K Locus	O Locus	731	731 + B1dep	B1dep	B1
1	NTUH-K2044 [43]	K1	ND	-	-	-	-
2	ABC429	K1	O1v2	-	-	-	-
3	BC-14-31	K1	O1v2	-	-	-	-
4	BC-15-45	K1	O1v2	-	-	-	-
5	BC-15-55	K1	O1v2	-	-	-	-
6	52145 [44]	K2	O1	-	+++	H	+++ *H* (high)
7	52145-Δ*wca*_K2_ [45]	K–	O1	+++ (high)	+++	-	-
8	52145-Δ*waaL* [46]	K2	O–	-	+++	H	+++ *H* (high)
9	52145-Δ*wca*_K2_Δ*waaL* [2]	K–	O–	+++ (high)	+++	-	+++ (low)
10	CIP 52.145	K2	O1	-	+++	H	+++ *H*
11	ABC83	K2	O1v1	-	H	H *	H
12	ABC127	K2	O1v1	-	-	- *	-
13	Kpn-ABC139	K2	O1v1	-	H	H *	H
14	ABC215	K2	O2v1	-	H	H *	H
15	ABC220	K2	O1v1	-	H	H *	H
16	ABC252	K2	O1v1	-	H	H *	H
17	ABC261	K2	O1v1	-	H	H *	H
18	ABC270	K2	O1v1	-	-	-/h *	-
19	BC14-298	K2	O1v2	-	-	-/H *	h
20	PFZ7	K2	O1v1	-	H	H *	H
21	PFZ10	K2	O1v1	-	H	H *	H
22	PFZ336S	K2	O1v1	-	H	H *	H
23	PFZ341	K2	O1v1	-	H	H *	H
24	PFZ594	K2	O1v1	-	-	- *	h
25	PFZ761	K2	O1v1	-	-	-	-
26	CIP 80.51	K3	ND	-	-	-	-
27	PFZ92S	K3	O1v2	-	-	-	-
28	ATCC 700603 [47]	K6	ND	-	-	-	-
29	CIP 52.207	K9	ND	-	-	-	-
30	ABC672	K9	O2v2	-	-	-	-
31	ABC111	K10	O3/O3a	-	-	-	-
32	ABC735	K10	O5	-	-	-	-
33	CIP 52.215	K11	O3	+	-	-	-
34	CIP 52.216	K12	O1	-	-	-	-
35	CIP 52.217	K13	ND	-	H	H	H
36	FD6-1	K15	O4	-	-	-	-
37	FD6-2	K15	O4	-	-	-	-
38	CIP 52.221	K17	ND	-	-	-	-
39	ABC621	K17	O1v1	-	-	-	-
40	ABC785	K17	O1v1	-	-	-	-
41	CIP 52.223	K19	ND	-	-	-	-
42	ABC39	K19	O2v2	-	-	-	-
43	ABC213	K19	O1v2	-	-	-	-
44	PFZ305S	K19	O102 (?)	-	-	-	-
45	PFZ401	K19	O1v2	-	-	-	-
46	PFZ410	K19	O1v2	-	-	-	-
47	PFZ412	K19	O1v2	-	-	-	-
48	PFZ604	K19	O1v2	-	-	-	-
49	CIP 52.224	K20	ND	+	-	-	-
50	ABC147	K20	O2v1	-	-	-	-
51	ABC484	K20	O2v1	-	-	-	-
52	KW140	K20	O2v1	-	-	-	-
53	KW141	K20	O2v1	-	-	-	-
54	KW144	K20	O2v1	-	-	-	-
55	CIP 52.225	K21	ND	+	+	-	-
56	ABC355	K21	O3b	-	-	-	-
57	ABC606	K21	O3b	+++ (high)	+++	-	-
58	53/3 [29]	K24	ND	-	-	-	-
59	CIP 52.229	K24	ND	-	-	-	-
60	ABC60	K24	O1v1	-	-	-	-
61	ABC360	K24	O2v1	+++ (high)	+++	-	-
62	ABC572	K24	O2v1	-	-	-	-
63	ABC593	K24	O2v1	-	H	H *	H
64	PFZ517	K25	O5	-	-	-	-
65	CIP 52.232	K27	O2	+	-	-	-
66	ABC776	K27	O4	-	-	-	-
67	PFZ334	K27	O4	+	-	-	-
68	PFZ560	K27	O4	-	-	-	-
69	CIP 52.235	K30	O1	-	-	-	-
70	ABC626	K30	O3/O3a (?)	-	-	-	-
71	KW150	K30	O1v2	-	-	-	-
72	KW154	K30	O1v2	-	-	-	-
73	CIP 53.8	K33	O3	+++ (high)	+++	-	-
74	KW1	K43	O2v1	-	-	-	-
75	KW2	K43	O2v1	-	-	-	-
76	CIP 53.23	K47	O1	-	-	-	-
77	PFZ679	K48	O1v1	-	-	-	-
78	PFZ682	K48	O1v1	-	-	-	-
79	PFZ687	K48	O1v1	-	-	-	-
80	ABC435	K51	O1v2	-	-	-	-
81	PFZ151S	K51	O1v2	-	-	-	-
82	PFZ155	K51	O1/O2v2	+	-	-	-
83	MGH 78578 [47]	K52	ND	-	-	-	-
84	ABC92	K54	O1v2	-	-	-	-
85	ABC196	K57	O3b	-	-	-	-
86	ABC493	K57	O2v1	-	-	-	-
87	ABC79	K62	O1v1	-	-	-	-
88	PFZ265	K62	O1v1	-	-	-	-
89	PFZ758	K62	O1v1	-	-	-	-
90	CIP 80.47	K64	ND	+	+	-	-
91	ABC217	K64	O2v1	-	-	-	-
92	ABC375	K64	O1v1	-	-	-	-
93	ABC669	K64	O2v1	-	-	-	-
94	ABC105	K74	O104	-	-	-	-
95	ABC587	K74	O104	-	-	-	-
96	ABC211	K106	O2v2	-	H	H *	H
97	ABC646	K107	O2v2	-	-	-	-
98	PFZ31	K110	O3b	-	-	-	-
99	PFZ673	K110	O3b	-	-	-	-
100	PFZ674	K110	O3b	-	-	-	-
101	PFZ542	K111	O3b	-	-	-	-
102	PFZ281	K112	O2v2 (?)	-	-	-	-
103	ABC135	K122	O2v2	-	-	-	-
104	ABC91	K136	O1v2	-	-	-	-
105	ABC718	K151	O4	-	-	-	-

**Table 2 microorganisms-11-00669-t002:** Comparison of Klebsiella phages from *Webervirus* genus, homologous to 731. Host and capsule specificity available in the literature is indicated. The phage genome similarities are presented in Figure 6, putative depolymerase homologies (for *orf22*) are in Figure 5. Order of the phages follows the multiple alignment by COBALT (Figure 5). Blank cells mean that the corresponding data is yet unknown. Accession numbers of the phages are listed in Appendix A.

Klebsiella Phage	Host Strain	Host Capsule Serotype	Other Susceptible Capsule Serotypes	Depoly-Merases	Reference
731	52145-Δ*wca*_K2_	no capsule	K21, K24, K33 (clear) K11, K20, K21, K27, K51, K64 (veiled)	*orf22*	[42], this study
GML-KpCol1					
vB_KpnS-VAC4	ATCC10031	K2	K30, K51, K112 (clear) K24, K38, K112 (veiled)		[36]
vB_KpnS-VAC7	ATCC10031	K2	K51, K112 (clear) K24, K30 (veiled)		[36]
vB_Kpn-VAC111					
vB_KpnS_15-38_KLPPOU149	15-38				
pK8	II-503				[59]
MezzoGao	ATCC51503				[60]
vB_KpnD_PeteCarol	ATCC10031	K2			[44]
PhiKpNIH-10	Phage Pharr (P1) and KpNIH-2 (P2) resistant MP103	no capsule			[61]
NJS2					
NJS1		no capsule	K47		[62]
Sweeny	1776c				[63]
vB_KpnS_Domnhall		K3			[64]
vB_KpnS_KingDDD		K3			[64]
vB_KpnS_SegesCirculi		>K3			[64]
PWKp14	Kpn32				[65]
Sushi	A1				[66]

## Data Availability

All data are presented in this manuscript in the main text and in the Appendix A.

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
