# Peer review of "Characterization of a Lytic Bacteriophage and Demonstration of Its Combined Lytic Effect with a K2 Depolymerase on the Hypervirulent Klebsiella pneumoniae Strain 52145"

_microorganisms, 2023, doi:10.3390/microorganisms11030669_

Round 1
Reviewer 1 Report
Dear authors,
I have found some minor mistakes pls correct them:
Line 6 - missing workplace
Line 13 - correct word
Line 324 - correct arrow in Table
Table 2 - correct trnaslation from HU to ENG

Reviewer 2 Report
Report
The present research microorganisms-2167944 titled: “Isolation and Characterization of a Novel Lytic Bacteriophage and Demonstration of its Combined Lytic Effect with a K2 Depolymerase on the Hypervirulent Klebsiella pneumoniae Strain 52145”was aimed at isolation and characterization of a bacteriophage against the nosocomial K. pneumoniae 52145 strain capsule defective mutant and studying the influence of combination of the recovered phage with K2 depolymerase. The topic is very interesting from the medical point of view however; there are some major and minor comments and suggestions that should be considered and fulfilled and these as follows:
Comments
1. Abstract: a. line 12, capsule defective should be changed to capsule-defective b. L126, 173 should be changed to phage-173. I recommend the author to use the standard method of writing the page code which in this case should be vB_Kp_173 c. L16, 173 should be changed to either to phage-173 or vB_Kp_173 (and the selected code should be used consistently in the whole manuscript). d. L17, 173 phage should be changed to either to phage-173 or vB_Kp_173 as previously mentioned and it should be used consistently in the whole manuscript. e. The authors should include some information about the genomic analysis of the recovered phage and putative tertiary structure of the ORF22 code for depolymerase. f. L18, B1dep abbreviation should be firstly described at the first mention. It should be inserted between brackets in L17 after mentioning “K2 depolymerase (B1dep). 2. Introduction section: a. The introduction section contains many citations of 63 references which are too many to be used in only one section. I recommend the author to reedit the introduction section be mainly focused on the core and aim of the study and delete all irrelevant information and significantly reduce the number of citations. b. L47, ST15 should be changed to sequence type-15 (ST15). c. L64, the sentence “Bacteria enveloped with polysaccharide capsules can be effectively controlled by phages “needs citation of the appropriate reference. 3. Material and methods. a. The author did not provide information about the storage and maintenance of the recovered phage 731 b. L89, sentence should not be started with number (100) so, I recommend to be change to a total of 100,…..etc). c. L90, 91, the authors should add a reference (including source or manufacture, City and country) of LB agar and LB broth. 4. Results: a. In Figure 1a and c, the lytic zones (plagues) is not clear. For example in Fig. 1a, there are lytic and clear zones and the author did not mention which one is the right. b. Figure 2 is not clear and not considered evidence about the synergistic effect. The authors should include controls (effect of each of the phage 731 and B1dep alone and their effects when they were combined. c. The author did not explain the major difference of phage 731 and phage 13 particularly on the level of depolymerase of each phage. d. The author did not deposit the assembles sequences/contigs in any of the public available databases such as NCBI or BV-BRC database or EMBL database in order to provide the accession code of the sequenced phage 731 on which all the nucleotides and respective ORFs (Including ORF22 coded for depolymerase) was provided to allow reviewing, monitoring and revising the phage under study. This is very important requirement and outcome of this study e. Results obtained were not satisfactory to reach the conclusion and to achieve the study goal particularly the effect of combinated lytic effects. f. Very similar results was previously published by the same authors in “Microorganism MDPI” Journal which reflects the lack of novelty conducted in this manuscript: Pertics BZ, Cox A, Nyúl A, Szamek N, Kovács T, Schneider G. Isolation and Characterization of a Novel Lytic Bacteriophage against the K2 Capsule-Expressing Hypervirulent Klebsiella pneumoniae Strain 52145, and Identification of Its Functional Depolymerase. Microorganisms. 2021 Mar 21;9(3):650. doi: 10.3390/microorganisms9030650. PMID: 33801047; PMCID: PMC8003838. 5. Results: The discussion section should be extensively modified in the light of the above comments and required modifications and updating as well as to highlight the novel findings of this manuscript as compared to the previously published article by the same authors.Therefore, and for the above mentioned remarks I advised a major revision of the respective manuscript in its current state taken into consideration the above comments and recommendations before being considered for publication

Reviewer 3 Report
The results presented are interesting.
However, the quality of the figures seriously compromises the clear understanding of the results. For example, in Figure 1 the burst size plots do not include the error bars. The burst size values (1000 and 10000) are unusually high. Are authors sure about these values? The quality of TEM image must be improved.
Genomic maps shown in Figure 4 seem to be duplicated.
Round 2
Reviewer 2 Report
The author did not deposit the assembled sequences/contigs in any of the publicly available databases such as NCBI or BV-BRC database or EMBL database in order to provide the accession code of the sequenced phage 731 on which all the nucleotides and respective ORFs (Including ORF22 coded for depolymerase) were provided to allow reviewing, monitoring and revising the phage under study. This is a very important requirement and outcome of this study
